# Characteristics of the Third COVID-19 Pandemic Wave with Special Focus on Socioeconomic Inequalities in Morbidity, Mortality and the Uptake of COVID-19 Vaccination in Hungary

**DOI:** 10.3390/jpm12030388

**Published:** 2022-03-03

**Authors:** Beatrix Oroszi, Attila Juhász, Csilla Nagy, Judit Krisztina Horváth, Krisztina Eszter Komlós, Gergő Túri, Martin McKee, Róza Ádány

**Affiliations:** 1Epidemiology and Surveillance Centre, Semmelweis University, 25. Üllői Street, 1085 Budapest, Hungary; oroszi.beatrix@semmelweis-univ.hu (B.O.); horvath.judit_krisztina@semmelweis-univ.hu (J.K.H.); komlos.krisztina_eszter@semmelweis-univ.hu (K.E.K.); turi.gergo@semmelweis-univ.hu (G.T.); 2Department of Public Health, Government Office of Capital City Budapest, 174. Váci Avenue, 1138 Budapest, Hungary; juhasz.attila@kmr.antsz.hu (A.J.); nagy.csilla@kmr.antsz.hu (C.N.); 3European Centre on Health of Societies in Transition (ECOHOST), London School of Hygiene and Tropical Medicine, Keppel Street, London WC1E 7HT, UK; 4MTA-DE-Public Health Research Group, Department of Public Health and Epidemiology, Faculty of Medicine, University of Debrecen, 26. Kassai Street, 4028 Debrecen, Hungary; 5Department of Public Health, Faculty of Medicine, Semmelweis University, 25. Üllői Street, 1085 Budapest, Hungary

**Keywords:** COVID-19, morbidity, mortality, excess mortality, vaccination coverage, deprivation, socioeconomic inequities, Roma

## Abstract

Governments are increasingly looking to vaccination to provide a path out of the COVID-19 pandemic. Hungary offers an example to investigate whether social inequalities compromise what a successful vaccine program can achieve. COVID-19 morbidity, mortality, and vaccination coverage were characterized by calculation of indirectly standardized ratios in the Hungarian population during the third pandemic wave at the level of municipalities, classified into deprivation quintiles. Then, their association with socioeconomic deprivation was assessed using ecological regression. Compared to the national average, people living in the most deprived municipalities had a 15–24% lower relative incidence of confirmed COVID-19 cases, but a 17–37% higher relative mortality and a 38% lower vaccination coverage. At an ecological level, COVID-19 mortality showed a strong positive association with deprivation and an inverse association with vaccination coverage (RR_Vaccination_ = 0.86 (0.75–0.98)), but the latter became non-significant after adjustment for deprivation (RR_Vaccination_ = 0.95 (0.84–1.09), RR_Deprivation_ = 1.10 (1.07–1.14)). Even what is widely viewed as one of the more successful vaccine roll outs was unable to close the gap in COVID-19 mortality during the third pandemic wave in Hungary. This is likely to be due to the challenges of reaching those living in the most deprived municipalities who experienced the highest mortality rates during the third wave.

## 1. Introduction

A government developing a response to a pandemic can employ a variety of approaches [1,2]. Some, such as restrictions on facilities where people mix or face mask mandates, seek to reduce transmission. Others, such as vaccine programmes, seek to increase immunity [3,4,5,6]. Additionally, others, such as financial support for those unable to work, offer help to those affected by the disease and necessary responses. The evidence, from empirical studies and models, points to a combination of measures as most likely to control disease spread and minimize collateral harms [7,8,9,10].

More restrictive measures to decrease population mixing and contact numbers often cause the biggest problems for those already disadvantaged, who cannot work from home, may work in the informal economy, live in overcrowded dwellings, and have few resources on which they can draw [11]. Those taking this perspective argue that relying on vaccines is less likely to exacerbate inequalities. However, inequities in the uptake of COVID-19 vaccination, which are reported from several countries, may jeopardize this goal [12,13].

During the 2020–2021 winter season, the new, fast-spreading Alpha variant appeared and gradually displaced the previous dominant variant in Europe [14,15]. In Hungary, the second wave and accompanying restrictions were not over when the even bigger third Alpha wave arrived at the beginning of 2021. This created an imperative to make rapid progress with the mass vaccination program, supplementing the four vaccines approved by the European Medicines Agency with Gamaleya Gam-COVID-Vac/Sputnik V and Beijing/Sinopharm BBIBP-CorV, granted emergency approval in Hungary. In Hungary, until the end of the third wave, 4,846,431 people became fully vaccinated (with a single dose of the Janssen vaccine or two doses of the others). A total of 61.3% received one of the four vaccines approved by the European Medicines Agency (41.7% Pfizer/BioNTech, 10.9% AstraZeneca, 6.6% Moderna and 2.1% Janssen), and 38.7% were vaccinated with Sputnik V (18.2%) or Sinopharm (20.5%) [16]. At the beginning of July 2021, Hungary had fully vaccinated a higher share of its population than any EU country except Malta, and higher than the UK, widely seen as an early leader [17]. Hungary is fortunate in having granular data on not only population characteristics but also vaccine uptake and COVID-19 epidemiology. We have previously taken advantage of this to analyze the association between these characteristics and patterns of COVID-19 during the second wave, finding that those in more deprived areas were less likely to be diagnosed with COVID-19 but more likely to die [18]. Here, we build on this earlier analysis by analyzing the epidemiology of the third wave, which occurred soon after the vaccination program had been rolled out.

Our study has several linked components. First, we describe the morbidity and mortality of the third pandemic wave, and we assess the spatial association between socioeconomic characteristics of the population and morbidity and mortality due to COVID-19. Then, we compare the second and third pandemic waves concerning the spatial distribution of confirmed COVID-19 cases and deaths. We then analyze the spatial distribution of vaccination coverage and its association with deprivation. Finally, we explore the interaction between vaccination coverage, mortality, and deprivation at the end of the third wave. In this way, we can look at the health consequences of relying primarily on mass vaccination, which had the benefit of speed but did not take account of the challenge of addressing inequalities. Unlike other studies that have used entire countries as the unit of analysis, this study exploits the substantial diversity within Hungary.

## 2. Methods

### 2.1. Data Collection

The Hungarian Notifiable Disease Surveillance System, operated by the National Public Health Center (NPHC), provided data on registered COVID-19 cases. Cases with laboratory confirmation (detection of SARS-CoV-2 by polymerase chain reaction (PCR) or SARS-CoV-2 antigen by a lateral flow test), irrespective of clinical signs and symptoms, were considered a confirmed case [19] and were involved in this study. The epidemic curve used the date of laboratory confirmation. Data on age, sex, municipality of residence, and outcome (alive/dead) were also available on an individual level.

Hungary is divided into 23 districts in Budapest and 197 in the 19 counties (the Hungarian names of these types of districts differ but can be considered the same for this analysis). Districts in counties, but not the capital, are further divided into a total of 3155 municipalities. Data on the share of Roma in the population were available only at the district level, while all other data were available at the municipality level (except in Budapest, where the lowest administrative level is the district).

Vaccination data (on doses administered in Hungary between week 52, 2020 and week 29, 2021) were provided by the National Health Insurance Fund of Hungary at the level of municipalities, by 5-year age groups, not broken down by sex.

The first and last days of the third pandemic wave in Hungary were defined as the days before and after the peak of the third wave, when the lowest daily case rate was recorded, in accordance with a previous study [18]. These days were 25 January 2021 and 4 July 2021.

Morbidity and mortality were defined as the incidence of COVID-19 cases or deaths, respectively, confirmed during the third wave. Cause of deaths was defined based on previous confirmation of COVID-19 by PCR or antigen test and on listing it as the underlying or a contributing cause of death.

Data on all-cause mortality for the third wave were obtained through the national Electronic Civil Registration System, which allowed us to investigate time trends by sex, age, and week of death. Excess mortality was calculated with reference to the years compared with the 2014–2019 mortality on data available as of 18 October 2021. The Central Office for Administrative and Electronic Public Services provided population data by sex and age at the municipality level for the same period.

As noted above, we have previously published data on the second pandemic wave (22 June 2020 and 24 January 2021) [18].

### 2.2. Deprivation Index

The area-based composite deprivation index [20], used successfully in several previous studies [18,21,22,23,24,25,26], provided information on socioeconomic deprivation at the municipality level. The index combined seven socioeconomic indicators (income, education level, unemployment rate, single-parent and large family households, housing density and car ownership) obtained from the databases of the Hungarian Central Statistical Office (2011 Census) and the Hungarian Tax and Financial Control Office (2011). Weights were estimated using principal component analysis. Higher values of the index correspond to lower socioeconomic status.

### 2.3. Statistical Analysis

Excess weekly mortality relative to the corresponding weeks in 2014–2019 were calculated for the third pandemic wave, from week 8 to week 18 of 2021, using the Rapid Inquiry Facility (RIF) [27] and applying age (5-year age groups) and sex-specific rates in the Hungarian population.

Consistent with our earlier study, we conducted a hierarchical Bayesian analysis [18,27,28]. The smoothed indirectly standardized ratios were estimated using the Integrated Nested Laplace Approximation (INLA) method [29] with expected cases adjusted for age (by 5-year age groups), sex and vaccination rates by age (again 5-year age groups). On the maps, areas were considered to have significantly high or low relative ratios if the estimated risk values differ from 1 with at least 80% probability [30].

Spatial scan statistics were applied to identify age, and sex-adjusted morbidity, mortality, and vaccination clusters with elevated or reduced ratios, confirming and complementing the results of disease mapping [27,31]. Fifty percent of the population at risk was defined as the maximum spatial cluster size.

We used the risk analysis module of RIF to assess the association between deprivation and morbidity, mortality (sex-specific) and vaccination coverage [27]. Municipalities were categorized into deprivation quintiles and indirect standardized ratios were calculated for each band. Chi-square tests for homogeneity and linearity trends were used to assess the overall association with deprivation.

Hungary, like other central European countries, has a large Roma population that experiences high levels of deprivation and worse health status and access to care. Thus, we looked in detail at municipalities in the 10 districts with the highest proportion of Roma in the population (27.9–39.0%), as identified in a previous nationwide survey by Pénzes et al. [32,33].

### 2.4. Shared Component Model

The spatial similarity of morbidity and mortality due to COVID-19 between the second and third pandemic waves was assessed using the shared component model. A “centered” version of the full Bayesian spatial shared component disease model proposed by Knorr-Held and Best [34] was applied to explore spatial risk patterns using conditional autoregression priors for the latent components to capture the spatial distribution of shared component [35]. The aim of our analysis was to identify the spatial pattern of the shared component, so we included only spatially structured random effects for the shared and time-specific latent components.

Expected case numbers were defined based on age- (by 5-year age groups) and sex-specific rates of the Hungarian population for the second and the third pandemic wave, respectively.

The models were fitted to the data using full Bayesian estimation and implemented in WinBUGS software [36]. Two different chains were run, and convergence was assessed using Gelman–Rubin’s convergence diagnostics [37]. The first 500,000 samples were discarded as burn-in and we ran a further 500,000 iterations, keeping every 50th, which were used in the calculation of the posterior estimates. The posterior probability of each area having an average risk above one (RR > 1) was computed as a measure of uncertainty and areas with high or low relative ratios were marked, where the estimated risk values were at least 80% likely to differ from 1 [30].

### 2.5. Ecological Regression

An ecological regression model was used to investigate simultaneously spatial relationships between COVID-19 vaccination, deprivation, and COVID-19 mortality. By extending the disease mapping models [28] with the spatial risk factor, we can take into account the assumed effect of residual spatial autocorrelation as an ‘unmeasured risk factor’. Ignoring this component may lead to biased estimates of regression coefficients, or underestimation of uncertainty by falsely estimating narrower confidence intervals [38].

The association between COVID-19-related mortality and vaccination coverage, controlling for deprivation as a confounder, was investigated at the municipality level by ecological regression using the INLA method [29]. The collinearity of the regression model was assessed by determining the variance inflation factor (VIF) [39]. The Deviance Information Criterion (DIC) and the Watanabe–Akaike Information Criterion (WAIC) were applied to select the best-fitting model from a variety of models [40,41].

## 3. Results

During the third pandemic wave, the cumulative number of reported confirmed COVID-19 cases was 447,966 (4529 cases per 100,000 population), while the reported number of deaths was 15,494 (crude case-fatality ratio 3.5%) (recorded and validated until 25 October 2021) (Figure 1). Although the duration of the third wave was shorter than that of the second wave, the cumulative number of cases reported was 25.8% higher than during the second wave (356,194 cases, 3601 cases per 100,000 population).

The daily number of newly registered cases started to increase during the last week of January 2021, followed by an increase in COVID-19-related deaths 3 weeks later (Figure 1). The peak of confirmed new cases was recorded on 25 March (11,266 or 114 per 100,000), and that of deaths (by date of death) on 1 April (289, or 3 per 100,000). At the peak of the third wave, the number of new cases (11,266) was 65.3% higher than at the peak of the second wave (6817).

The highest crude case-fatality ratio was observed in the 80+ years age group (30.9%), followed by the 65–79 years (12.5%) and the 50–64 years age group (2.7%). A total of five deaths were reported among those under 18 years (two under 1 year, and three aged between 14 and 16 years). Among deaths, 76.4% (11,830 deaths) were in the 65+ years age group.

Of the 447,966 cases, 52.9% (236,797) were female and 47.1% (211,169) male. Confirmed cases among women were higher (4626 vs. 4426 per 100,000 population); men were 0.96 (95% CI 0.95 to 0.96) as likely to be ascertained as a COVID-19 case. However, men had 1.2 (95% CI 1.15 to 1.23) times the risk of death (171 cases vs. 143 cases per 100,000).

During the third COVID-19 pandemic wave, excess mortality from all causes started to increase steeply after week 8, 2021. Between week 8 and 19, significant excess deaths were found in all age groups (excess mortality_0–X years_: 1.32 (CI: 1.33–1.34), excess number of death_0–X years_: 10,633 (CI: 10,326–10,937); excess mortality_50–64 years_: 1.40 (CI: 1.39–1.42), excess number of death_50–64 years_: 2093 (CI: 1970–2212); excess mortality_65–X years_: 1.32 (CI: 1.30–1.33), excess number of death_65–X years_: 7994 (CI: 7720–8265)) (Figure 2A–C). In total, excess mortality was 32% higher than the average weekly mortality for the period 2014–2019 for all ages, 40% higher for 50–64 years, and 32% higher for 65 years or over (Figure 2A–C).

The relative excess mortality was the highest on week 13 in all age groups (for age 0–X years 1.69 (1.64–1.74)) and for 50–64 years, 1.77 (1.66–1.90), as well as in the 65 years or over age groups (1.65 (1.59–1.71)). Compared to the 2014–2019 averages in corresponding weeks, excess mortality was 69%, 77% and 65% higher, respectively (Figure 2A–C).

The vaccination campaign started on week 52, 2020 with the Pfizer-BioNTech/BNT162b2 COVID-19 vaccine. The Moderna/mRNA-1273 COVID-19 vaccine became available from week 2, while the Oxford/AstraZeneca ChAdOx1-S and Gamaleya Gam-COVID-Vac/Sputnik V from week 6, the Beijing/Sinopharm BBIBP-CorV from week 8, and the Janssen/Ad26.COV2.S COVID-19 vaccine (also referred to as the Johnson & Johnson vaccine) from week 18, 2021.

The vaccination campaign ran in parallel with the third pandemic wave in Hungary. At the peak of the epidemic (week 12, 2021), the uptake of the first dose was only 15.0% in those aged 18 and above and the proportion of fully vaccinated (who had received the second dose) reached 7.2%. By the end of the second wave (week 26, 2021), 59.3% of this age group was fully vaccinated, with 5.1% only partially vaccinated (Figure 3).

When we look at the spatial distribution of COVID-19 vaccination uptake by the end of the third wave, we can see significantly higher vaccination coverage in the western part of the capital, Budapest, in county towns and in larger cities. Significantly lower relative vaccination coverage was observed in the more sparsely populated areas of north-eastern and south-western Hungary (Figure 4A,B).

The risk analysis showed a significant inverse relationship between vaccination coverage and deprivation (χ^2^Homogeneity = 95,849.26, *p* = 0; χ^2^Linearity = 81,178.63, *p* = 0) (Figure 5C). Areas of highest deprivation (V. quintile) experienced the lowest relative vaccination ratio (Relative Vaccination Ratio_both sexes, V_._quint_.: 0.618 (CI: 0.616–0.620)) (Figure 4B,C, Table 1). In contrast, in the least deprived (I. quintile) vaccination coverage was 9% higher (Table 1). In the municipalities in the 10 districts with the highest proportion of Roma population, the vaccination rate was only 55% of the national rate (Relative Vaccination Ratio_both sexes, Highest proportion of Roma population_.: 0.550 (CI: 0.560–0.640) (Table 1).

Spatial inequalities of deprivation, incidence ratio, and relative mortality due to COVID-19 at the municipality level are illustrated in Figure 5.

The most deprived areas, measured by DI, are in the north-eastern and south-western parts of Hungary, and the least-deprived municipalities are in the north-western parts, as well as Budapest and neighboring areas (Figure 5A). The third wave of the COVID-19 pandemic did not affect these different areas in the same way.

A high incidence ratio of registered cases was recorded in the north-western and mid-west parts, as well as in some parts in the northeast and south (Figure 5B). High mortality ratios were recorded on the north-eastern border and in the central and mid-west parts (Figure 5C). Areas of lower incidence and lower mortality risk were concentrated in the middle of the eastern part and southwestern border (Figure 5B,C).

The shared component analysis revealed similarities in spatial patterns of incidence and mortality of the second and third pandemic wave (Figure 5D,E). Significantly higher incidence ratios were observed in both waves in the north-western quarter of Hungary, including the capital city (Figure 5D), while higher mortality risks were seen in the north-eastern, north-central, central, and south-western parts of the country (Figure 5E).

The risk analysis found a significant inverse relationship between relative incidence of confirmed cases (Males: χ^2^Homogeneity = 1575.8, *p* = 0; χ^2^Linearity = 1119.51, *p* = 0; Females: χ^2^Homogeneity = 603.72, *p* = 0; χ^2^Linearity = 341.26, *p* = 0), and strong positive association between the relative mortality and deprivation (Males: χ^2^Homogeneity = 82.15, *p* = 0; χ^2^Linearity = 72.48, *p* = 0; Females: χ^2^Homogeneity = 115.36, *p* = 0; χ^2^Linearity = 111.95, *p* = 0) (Figure 6).

Areas with highest deprivation (V. quintile) had significantly lower relative incidence ratios of confirmed cases (relative incidence ratio males, V. quint.: 0.76 (CI: 0.74–0.77); relative incidence ratio females, V. quint.: 0.85 (CI: 0.83–0.86)) for both sexes (Figure 5A, Table 2). However, those people experienced the highest mortality (Relative Mortality_males, V_. _quint_.: 1.17 (CI: 1.07–1.26); Relative Mortality_females, V_. _quint_.: 1.37 (CI: 1.26–1.48)) (Figure 5B, Table 2). Overall, in quintile V (most deprived), the relative incidence was 15–24% lower, while the relative mortality was 17–37% higher than the national average (Table 2).

In municipalities of the 10 districts with the highest representation of Roma population, the mortality rate was 27% higher for men and 53% higher for women than the national rate (Relative Mortality_males, Highest representation of Roma pop_.: 1.27 (CI: 1.12–1.45); Relative Mortality_females, Highest representation of Roma pop_.: 1.53 (CI: 1.35–1.73)) (Figure 5C, Table 2).

In contrast, in the least deprived quintile, the incidence was only 3% higher than the national average for males, and it was similar to the national average for women, but mortality was almost 20% lower than the national average in both sexes.

Relative risk of morbidity and mortality due to COVID-19 by DI quintiles during the third pandemic wave in Hungary. The ecological regression showed an inverse association between vaccination and mortality (RR_Vaccination_ = 0.86 (0.75–0.98)), while deprivation had a strong positive association with mortality. However, the former became non-significant after adjusting for deprivation (RR_Vaccination_ = 0.95 (0.84–1.09), RR_Deprivation_ = 1.10 (1.07–1.14)).

## 4. Discussion

Despite having a complex package of nonpharmacological measures in place and a rapidly expanding vaccination campaign, one of the fastest in the EU with 24.4% of the population receiving at least one dose of COVID-19 vaccine by the time the third pandemic wave peaked and 52.5% fully vaccinated when it ended, Hungary was unable to contain the third pandemic wave, caused by the more virulent and faster spreading SARS-CoV-2 Alpha variant. Thus, the third wave exceeded the size of the second wave in terms of both the number of cases registered and the number of COVID-19-related deaths [42,43]. In addition, the premature excess mortality for the 50–64-year-old population was markedly higher for the third wave than in the second wave [18].

Hungary’s experience is a reminder of the need to look beyond the aggregate data. Although the vaccination campaign started before the third wave, differences in vaccine acceptance soon emerged, initially unnoticed. Less deprived, mainly urban areas achieved higher coverage so, by the end of the third wave, protection was strongly socially patterned. This is consistent with experience elsewhere, in the UK and the United States [12,13,44,45]. As expected, coverage was especially low in areas with the highest proportion of Roma population, even more so than in the other most deprived areas.

To our knowledge, our study is one of the first to investigate the health impact of a COVID-19 vaccination program within a country during a pandemic wave, incorporating data on both socioeconomic inequalities and vaccination. As expected, we found an inverse association between vaccination and mortality, with the latter, again, as expected, being strongly associated with deprivation. After adjusting for deprivation as a confounder, there was no evidence of an association between municipal level vaccination coverage and the spatial pattern of mortality. This seems counterintuitive, given the clear evidence of the effectiveness of the vaccines.

Concerning the fact that 38.7% of the population fully vaccinated had received a vaccine with no European Medicines Agency authorization but emergency approval in Hungary (Sputnik V or Sinopharm), it is reasonable to suppose that the type of vaccine administered might influence our results. In the present study, this question was not investigated, but it seems very likely that the distribution of different vaccines administered did not influence our results significantly, which is supported by the results of a nationwide, retrospective observational study (conducted between 22 January and 10 June 2021 in Hungary), which showed similarly high effectiveness of the different two-dose vaccines used in Hungary. Regarding morbidity among fully vaccinated people, incidence rates varied in a relatively narrow range—between 0.04 (Sputnik V) and 0.6 (Sinopharm) per 100,000 person days—in the relevant fully vaccinated populations [46]. In line with the Hungarian data in a community-based, retrospective, observational study carried out in the United Arab Emirates, it was shown that Sinopharm vaccine effectiveness in fully vaccinated individuals was 80%, 92%, and 97% in preventing COVID-19-related hospital admissions, critical care admissions, and death, respectively [47]. A recent narrative review based on the comparative analysis of the characteristics, adverse events, efficacy, effectiveness, and impact of 19 variants (including Sputnik V and Sinopharm) of COVID-19 vaccines also concludes that “all vaccines appear to be safe and effective tools to prevent severe COVID-19, hospitalization, and death”, although BNT162b2, mRNA-1273 and Sputnik V after two doses had the highest efficacy (>90%). As a limitation of this narrative review, it was mentioned that large observational studies were lacking for several authorized vaccines and, even if they were carried out, are subject to bias when assessing effectiveness, such as misclassification from diagnostic errors, imbalances in socioeconomic status, exposure risk, healthcare-seeking behaviors, or immunity status between vaccinated and unvaccinated groups [48].

However, there are several plausible explanations. First, although the Hungarian vaccine roll out was fast, the spread of the pandemic was even faster, and it took time to reach high numbers of people who had two doses. Secondly, and assuming patterns observed elsewhere apply equally to Hungary, those individuals and households that are most disadvantaged within a given area are most likely to become infected and severely ill and be unvaccinated.

The deprivation index is a multidimensional index that considers and combines several socioeconomic factors, including educational level. Several studies have confirmed that education significantly influences the willingness to accept vaccination, as those with lower education are more likely to refuse COVID-19 vaccination [49,50,51,52,53]. Since education is one of the most important (if not the most important) determinants of the deprivation index [20], it is assumed that higher education and the related higher willingness to accept vaccination may be a decisive factor for the higher vaccination coverage identified in more advantaged socioeconomic areas. In addition, in a recent study covering 41 countries, it was clearly shown that low levels of educational attainment (in 66% of countries) and low income (80%) were also positively correlated with non-compliance with the recommendations/restrictions [54].

This study adds to the literature by describing the epidemiology of COVID-19 incidence at a granular level within a country where it can be related to the socioeconomic situation. It is striking that both waves significantly affected the north-western quarter of the country and the capital, as measured by recorded COVID-19 cases, but these areas had a relative mortality lower than the national average. In contrast, the north-eastern parts of the country were hit hard in terms of mortality in both waves, but this would not have been expected on the basis of recorded case numbers. Thus, the highest morbidity was in the least deprived areas, but the highest mortality was in the most deprived areas. This was also found in the second wave [18]. Additionally, as seen in the second wave, areas with the highest proportion of Roma population had mortality rates considerably higher than the national rate, more than could be explained by the aggregate measure of deprivation.

The persistence of these patterns points to the role of unmeasured factors, likely related to inequalities in testing, contact tracing, access to health care, and in risk-avoiding behaviors.

Concerning the districts with the highest representation of Roma population, a higher relative mortality ratio was identified for Roma females than for Roma males, but because of the different sex-specific reference rates, the relative mortality ratio for females is not directly comparable for males. However, considering 95% confidence intervals around the point estimates, the magnitude of the deviation from the sex-specific references, which indicates the degree of inequality, does not differ significantly between Roma men and women.

This study is subject to several limitations. No spatial analysis can capture all relevant individual risk factors, such as underlying health conditions. Ascertainment of cases is influenced by access to testing, willingness to test and to be tested, and possibly enhanced surveillance of certain groups, such as those with chronic illness or a higher threshold for testing in those presumed to be at less risk having been vaccinated. Our findings point to a persistence of undertesting among those living in the most disadvantaged areas in both pandemic waves [18]. The lower incidence rates in the more deprived areas do not necessarily mean that there are fewer COVID-19 infections, but they may indicate underdiagnosis due to less testing. This hypothesis cannot be verified, because data about the number of COVID-19 tests performed at the municipality level are not available in Hungary. COVID-19 testing is available both free of charge and for a fee in Hungary. However, access and availability to free testing and, in particular, fee-based testing may have been influenced and restricted by socioeconomic status. Ascertainment of COVID-related death can be also biased due to geographical differences in the determination of causes of death but, given the duration of the pandemic, the importance of these problems may have diminished.

Assuming that biases and uncontrolled confounding have remained constant, our results concerning COVID-19 morbidity and mortality can be compared over time. Vaccination data may be incomplete, which might cause some underestimation of the coverage data over time.

Another limitation of the study is the limited timeliness of the deprivation index. Most of the indicators used for the index are only available from the 2011 census data and, therefore, as for all such area-based socioeconomic indicators, the accuracy of the index diminishes over time. It should be noted, however, that the spatial distribution of the indices based on the 2001 and 2011 census data, compiled with the same methodology (although not directly comparable in terms of values), did not show any significant, marked change and the deprived areas showed a nearly identical spatial distribution within Hungary, which suggests that the socioeconomic characteristic at the territorial level is not changing, or if so, only very slowly [18,20].

In conclusion, despite the availability of several effective vaccines, the Hungarian vaccination program faced difficulties in reaching those in the most disadvantaged areas. The ecological regression showed that the uneven coverage by the vaccination program was inadequate to overcome this disadvantage and more effort was needed to reach certain groups, who were likely at greatest risk of COVID-19. We have safe and effective vaccines, but they can only work if they reach those at greatest need.

## Figures and Tables

**Figure 1 jpm-12-00388-f001:**
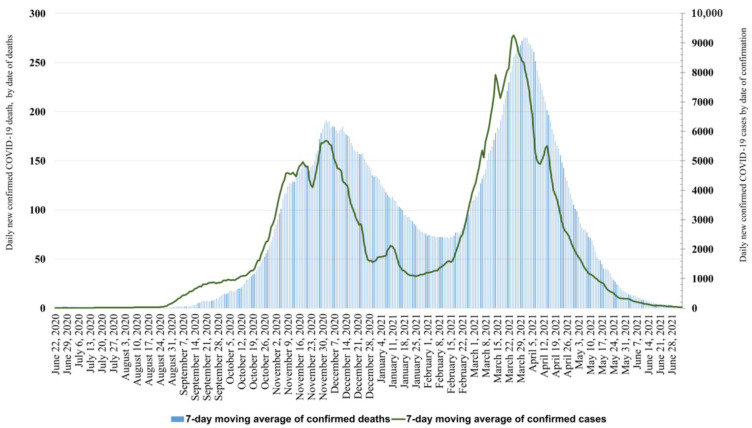
Daily number of confirmed COVID-19 cases (by date of confirmation) and deaths (by date of death) during the second (22 June 2020–24 January 2021) and the third (25 January 2021–4 July 2021) pandemic waves in Hungary.

**Figure 2 jpm-12-00388-f002:**
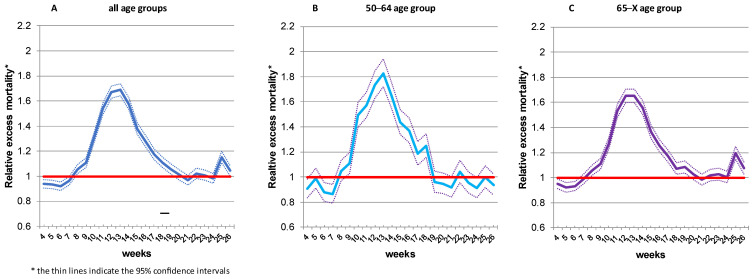
Relative excess mortality due to all causes of death during the third COVID-19 pandemic wave, compared to the average weekly mortality for the period 2014–2019 for all age groups (**A**), for 50–64 age groups (**B**) and for 65–X age groups (**C**) in Hungary.

**Figure 3 jpm-12-00388-f003:**
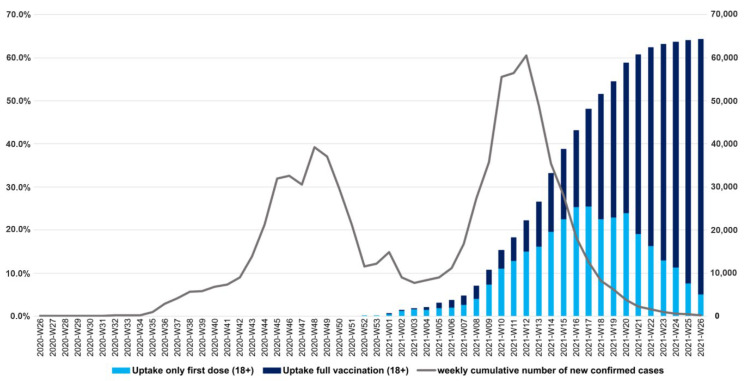
Cumulative uptake (%) of the first dose and full vaccination in Hungary and weekly number of confirmed cases between week 26, 2020 and week 26, 2021.

**Figure 4 jpm-12-00388-f004:**
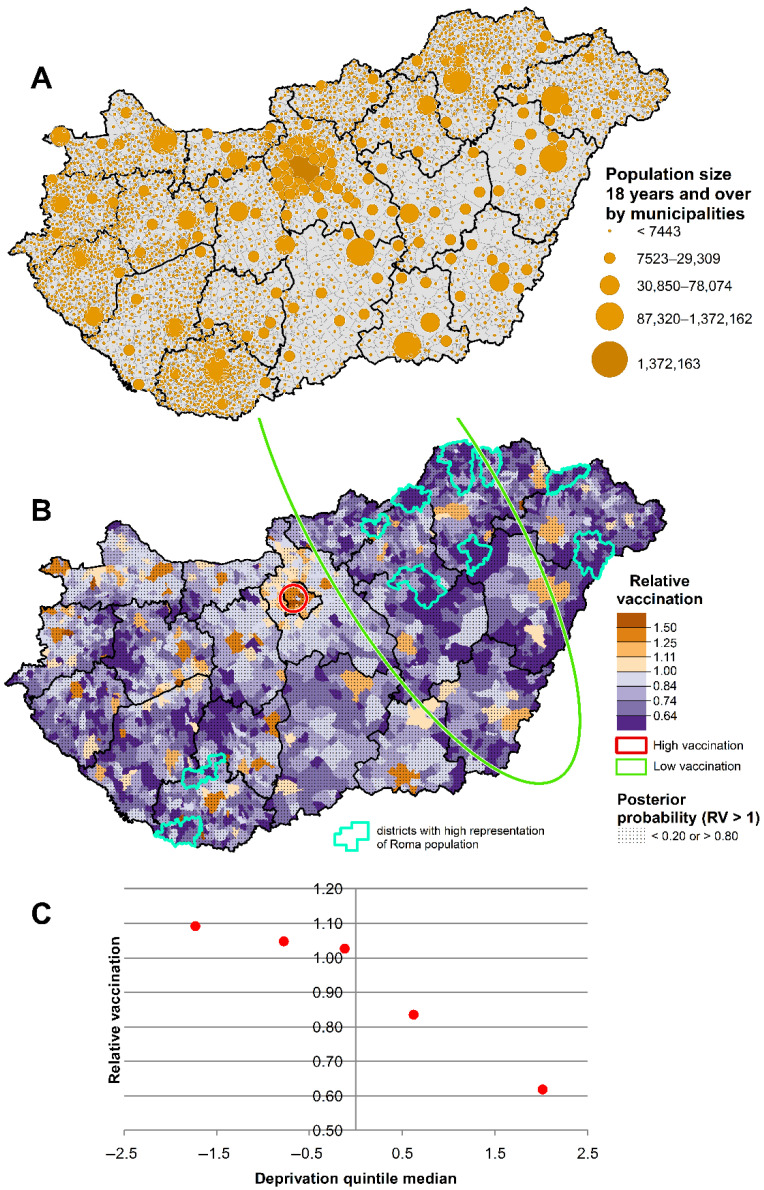
The spatial inequalities in population density (**A**), relative vaccination coverage against COVID-19 (**B**) and the relationship between the deprivation and vaccination coverage against COVID-19 (**C**) by Deprivation Index quintiles during the third pandemic wave in Hungary.

**Figure 5 jpm-12-00388-f005:**
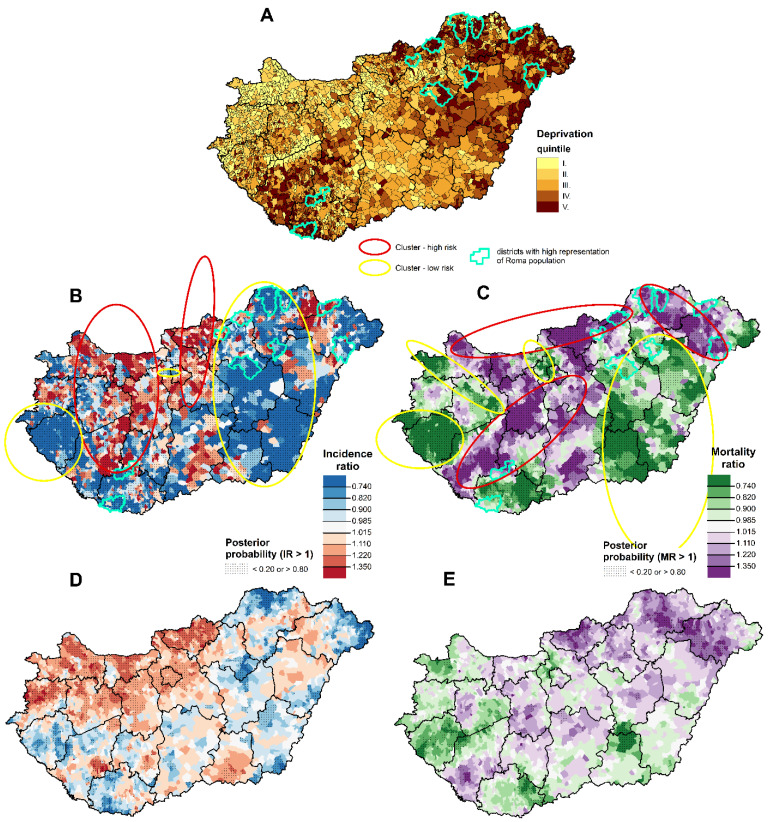
The spatial distribution of deprivation (**A**), incidence ratio of confirmed cases (**B**), relative mortality (**C**) due to COVID-19, during the third pandemic wave and the shared component of the second and third pandemic wave incidence (**D**) and mortality (**E**) in Hungary.

**Figure 6 jpm-12-00388-f006:**
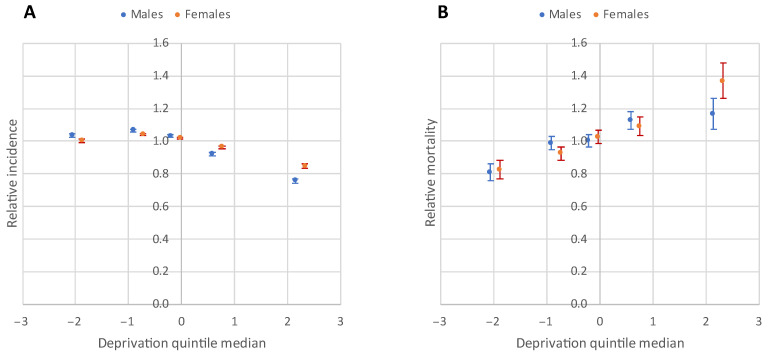
Relationship between the deprivation and relative incidence of confirmed cases (**A**), relative mortality (**B**) due to COVID-19 by DI quintile and sex during the third pandemic wave in Hungary.

**Table 1 jpm-12-00388-t001:** Relative vaccination coverage ratio against COVID-19 by Deprivation Index quintiles and in the districts with the highest representation of Roma during the third pandemic wave in Hungary.

DI Quintiles	Relative Vaccination Coverage Ratio
I. (least deprived)	1.092 [1.090–1.095]
II.	1.047 [1.045–1.049]
III.	1.026 [1.024–1.028]
IV.	0.834 [0.833–0.836]
V. (most deprived)	0.618 [0.616–0.620]
Districts with highest representation of Roma population	0.550 [0.560–0.640]

**Table 2 jpm-12-00388-t002:** Relative risk of morbidity and mortality due to COVID-19 by DI quintiles during the third pandemic wave in Hungary.

DI Quintiles	Confirmed Cases	Relative Incidence Ratio	Death Cases	Relative Mortality Ratio
Males				
I. (least deprived)	31,150	1.03 [1.02–1.05]	1003	0.81 [0.76–0.86]
II.	63,455	1.07 [1.06–1.08]	2365	0.99 [0.95–1.03]
III.	68,449	1.03 [1.02–1.04]	2581	1.00 [0.96–1.04]
IV.	35,831	0.92 [0.91–0.93]	1618	1.13 [1.07–1.18]
V. (most deprived)	12,284	0.76 [0.74–0.77]	587	1.17 [1.07–1.26]
Districts with highest representation of Roma population	4526	0.81 [0.79–0.83]	230	1.27 [1.12–1.45]
Females				
I. (least deprived)	33,719	1.00 [0.99–1.01]	852	0.82 [0.77–0.88]
II.	70,667	1.04 [1.03–1.05]	2013	0.93 [0.89–0.97]
III.	76,691	1.02 [1.01–1.03]	2422	1.03 [0.99–1.07]
IV.	41,243	0.96 [0.95–0.97]	1440	1.09 [1.04–1.15]
V. (most deprived)	14,477	0.85 [0.83–0.86]	613	1.37 [1.26–1.48]
Districts with highest representation of Roma population	5550	0.92 [0.90–0.95]	255	1.53 [1.35–1.73]

## Data Availability

All relevant data may be obtained from the corresponding author upon a reasonable request and prior permission of the study funder.

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
