# Peer review of "Characteristics of the Third COVID-19 Pandemic Wave with Special Focus on Socioeconomic Inequalities in Morbidity, Mortality and the Uptake of COVID-19 Vaccination in Hungary"

_jpm, 2022, doi:10.3390/jpm12030388_

Round 1
Reviewer 1 Report
The authors present a study on the third COVID-19 pandemic wave in Hungary (2021 Q1+Q2), and focus on socioeconomic inequalities in morbidity, mortality and the uptake of vaccines. The study seems scientifically sound and the results are described in a clear manner.
Some questions and comments:
- line 56-59: which vaccines were administered in what proportion, and does this influence your results in any way?
- line 65: "finding that those in more deprived areas were less likely to be diagnosed with COVID-19 but more likely to die" -> Is this underdiagnosis due to less COVID-19 or due to less testing? Is testing free in Hungary? If there any costs, this might have a negative effect on the number of diagnoses in more deprived areas.
- Line 186-191: There are, on average, 9 days between infection and hospitalization and 13 days between hospitalization and death (https://journals.plos.org/plosone/article?id=10.1371/journal.pone.0257978). So indeed the 3 weeks make sense. But do you have any suggestion why there are only six days between the peaks of March 25 and April 1?
- Line 286: Alpha -> Shouldn't this be Delta?
Author Response
Response to Reviewer 1
(jpm-1605375 )
Thank the Reviewer for taking the time to assess our manuscript and giving valuable suggestions. We have addressed all the concerns raised by him/her. Here are the comments and the answers to each one.
“line 56-59: which vaccines were administered in what proportion, and does this influence your results in any way?”
We agree with the Reviewer that further elaboration on this point would be helpful. Thus, the following sentences have been added to the manuscript,
- to the Introduction section:
“In Hungary until the end of the third wave 4 846 431 people became fully vaccinated (with a single dose of the Janssen vaccine or two doses of the others). 61.3% received one of the four vaccines approved by the European Medicines Agency (41.7% Pfizer/BioNTech, 10.9% AstraZeneca, 6.6% Moderna and 2.1% Janssen), and 38.7% were vaccinated with Sputnik V (18.2%) or Sinopharm (20.5%). [The source of data is cited as reference 16: ECDC, Data on COVID-19 vaccination in the EU/EEA, Accessed 24 Feb 2022, https://www.ecdc.europa.eu/en/publications-data/data-covid-19-vaccination-eu-eea]. (Lines 60-65 in the revised document.)
- to the Discussion section:
Concerning the fact that 38.7% of the population fully vaccinated had received a vaccine with no EMA authorization but emergency approval in Hungary (Sputnik V or Sinopharm), it is reasonable to suppose that the type of vaccine administered might influence our results. In the present study, this question was not investigated, but it seems very likely that the distribution of different vaccines administered did not influence our results significantly which is supported by the results of a nation-wide, retrospective, observational study (conducted between 22 January and 10 June 2021 in Hungary) which showed similarly high effectiveness of the different two-dose vaccines used in Hungary. Regarding morbidity among fully vaccinated people, incidence rates varied in a relatively narrow range - between 0.04 (Sputnik V) and 0.6 (Sinopharm) per 100,000 person-days - in the relevant fully vaccinated populations [46]. In line with the Hungarian data in a community-based, retrospective, observational study carried out in the United Arab Emirates it was showed that Sinopharm vaccine effectiveness in fully vaccinated individuals was 80%, 92%, and 97% in preventing COVID-19-related hospital admissions, critical care admissions, and death, respectively [47].” A recent narrative review based on the comparative analysis of the characteristics, adverse events, efficacy, effectiveness, and impact of 19 variants (including Sputnik V and Sinopharm) of COVID-19 vaccines also concludes that “all vaccines appear to be safe and effective tools to prevent severe COVID-19, hospitalization, and death”, although BNT162b2, mRNA-1273 and Sputnik V after two doses had the highest efficacy (>90%). As a limitation of this narrative review is mentioned that large observational studies were lacking for several authorized vaccines and even if they were carried out are subject to bias when assessing effectiveness, such as misclassification from diagnostic errors, imbalances in socioeconomic status, exposure risk, healthcare-seeking behaviors, or immunity status between vaccinated and unvaccinated groups [48]. (Lines 344-368 in the revised document.)
References added:
- Vokó, Z.; Kiss, Z.; Surján, G.; Surján, O.; Barcza, Z.; Pályi, B.; Formanek-Balku, E.; Molnár, G.A.; Herczeg, R.; Gyenesei, A.; et al. Nationwide Effectiveness of Five SARS-CoV-2 Vaccines in Hungary—the HUN-VE Study. Clinical Microbiology and Infection 2021, 0, doi:10.1016/j.cmi.2021.11.011.
- Ismail AlHosani, F.; Eduardo Stanciole, A.; Aden, B.; Timoshkin, A.; Najim, O.; Abbas Zaher, W.; AlSayedsaleh AlDhaheri, F.; Al Mazrouie, S.; Rizvi, T.A.; Mustafa, F. Impact of the Sinopharm’s BBIBP-CorV Vaccine in Preventing Hospital Admissions and Death in Infected Vaccinees: Results from a Retrospective Study in the Emirate of Abu Dhabi, United Arab Emirates (UAE). Vaccine 2022, S0264-410X(22)00174-8, doi:10.1016/j.vaccine.2022.02.039.
- Fiolet, T.; Kherabi, Y.; MacDonald, C.-J.; Ghosn, J.; Peiffer-Smadja, N. Comparing COVID-19 Vaccines for Their Characteristics, Efficacy and Effectiveness against SARS-CoV-2 and Variants of Concern: A Narrative Review. Clin Microbiol Infect 2022, 28, 202–221, doi:10.1016/j.cmi.2021.10.005.
“line 65: "finding that those in more deprived areas were less likely to be diagnosed with COVID-19 but more likely to die" -> Is this underdiagnosis due to less COVID-19 or due to less testing? Is testing free in Hungary? If there any costs, this might have a negative effect on the number of diagnoses in more deprived areas.”
We thank the Reviewer for pointing these issues out.
As for your question, we think that underdiagnosis is mainly due to less COVID-19 testing, however, data about the number of COVID-19 tests performed at the municipality level is not available in Hungary. That’s why we concluded it in lines 354-356 of the Discussion section: ”Our findings point to a probable persistence of undertesting among those living in the most disadvantaged areas in both pandemic waves.”
The following sentences have been added to the discussion section:
“The lower incidence rates in the more deprived areas do not necessarily mean that there are fewer COVID-19 infections, but they may indicate underdiagnosis due to less testing. This hypothesis cannot be verified, because data about the number of COVID-19 tests performed at the municipality level is not available in Hungary. COVID-19 testing is available both free of charge and for a fee in Hungary. However, access and availability to free testing and in particular fee-based testing may have been influenced and restricted by socio-economic status.” (Lines 413-419 in the revised document.)
“line 186-191: There are, on average, 9 days between infection and hospitalization and 13 days between hospitalization and death (https://journals.plos.org/plosone/article?id=10.1371/journal.pone.0257978). So indeed the 3 weeks make sense. But do you have any suggestion why there are only six days between the peaks of March 25 and April 1?”
The daily number of newly registered cases started to increase during the last week of January 2021, followed by an increase of COVID-19 related deaths 3 weeks later. And later, indeed, the peak of confirmed new cases (based on the date of confirmation) was recorded only 1 week before the peak of deaths (by date of death). We assume a complex background and suppose that the followings may all have contributed to this phenomenon:
- The epicurve of confirmed cases (7-day moving average) is based on the date of confirmation of COVID-19. Thus, there may be some delay between the manifestation of the infection (onset date of first symptoms) and the confirmation date used in our analysis. As the pandemic wave progressed and testing capacities came under increasing pressure, the delay between the onset of symptoms and the time of diagnosis may have increased, and consequently, the interval between diagnosis and death may have decreased.
- The general health status of Hungarian patients may be worse than that of those people involved in the analysis taking place in the United Kingdom (https://journals.plos.org/plosone/article?id=10.1371/journal.pone.0257978).
- The national epicurve is added up by the sum of the smaller regional, municipal epidemic curves. There are well-known geographical inequalities in the health status of the population and thus in their vulnerability. In high-vulnerability populations, the epidemic could have started earlier in time, which could bring the peak of the national mortality curve forward.
“line 286: Alpha -> Shouldn’t this be Delta?”
We thank the Reviewer for pointing out our mistake in using the proper variant name. After checking the use of the term “Alpha” throughout the manuscript, we can confirm that “Alpha” is properly used in line 310, but not in line 55 along with the misuse of “Delta” in line 56. The sentence in lines 54-56 of the Introduction has been modified accordingly:
“In Hungary, the second wave and accompanying restrictions were not over when the even bigger third, Alpha, wave arrived at the beginning of 2021.” (Lines 55-57 in the revised version.)
Reviewer 2 Report
The data presented by the authors, although interesting, do not clearly show what may be caused by the differences in vaccination coverage, morbidity and deaths. The authors do not try to answer the question whether, for example, education may influence the willingness to accept the vaccine or to comply with the restrictions. At least a few sentences on this subject, based on the data presented, should appear in Conclusion.
In the introduction, the authors describe their research and subsequent steps taken. They should, however, describe in more detail what exactly was compared in the subsequent phases of the pandemic (lines 69-70). Failure to list individual elements raises the question of why the comparison was made at the very beginning and not all possible data presented?
Line 118 - 2011 data are out of date, so why did the authors relied on them?
Lines 196-197 - are the 5 cases described below 1 year and between 14-16? If so, then this detail should be in parentheses. Otherwise the information is incomprehensible. Currently, it shows that, in addition to 5 people under the age of 18, also 2 died under 1 and 3 between 14-16.
Information about a higher incidence rate among the Roma community is predictable. And it certainly results, among other things, from the way of life of this community. It is interesting, however, that the mortality rate among women in this social group is higher. However, there is no information to what extent these differences are statistically significant.
Author Response
Response to Reviewer 2
(jpm-1605375 )
Thank the Reviewer for taking the time to assess our manuscript and giving valuable suggestions. We have addressed all the concerns raised by him/her. Here are the comments and the answers to each one.
“The data presented by the authors, although interesting, do not clearly show what may be caused by the differences in vaccination coverage, morbidity and deaths. The authors do not try to answer the question whether, for example, education may influence the willingness to accept the vaccine or to comply with the restrictions. At least a few sentences on this subject, based on the data presented, should appear in Conclusion.”
We agree with the Reviewer and the following section has been added to the Discussion section:
The deprivation index is a multidimensional index that considers and combines several socio-economic factors, including educational level. Several studies have confirmed that education significantly influences the willingness to accept vaccination, as those with lower education are more likely to refuse COVID-19 vaccination [49–53]. Since education is one of the most important (if not the most important) determinants of the deprivation index [20], it is assumed that higher education and the related higher willingness to accept vaccination may be a decisive factor for the higher vaccination coverage identified in more advantaged socio-economic areas. In addition, in a recent study covering 41 countries, it was clearly shown that low levels of educational attainment (in 66% of countries) and low income (80%) were also positively correlated with non-compliance with the recommendations/restrictions [54]. (Lines 375-385 in the revised document.)
Added pieces of literature:
20. Juhász, A.; Nagy, C.; Páldy, A.; Beale, L. Development of a Deprivation Index and Its Relation to Premature Mortality Due to Diseases of the Circulatory System in Hungary, 1998-2004. Soc Sci Med 2010, 70, 1342–1349, doi:10.1016/j.socscimed.2010.01.024.
49. Kessels, R.; Luyten, J.; Tubeuf, S. Willingness to Get Vaccinated against Covid-19 and Attitudes toward Vaccination in General. Vaccine 2021, 39, 4716–4722, doi:10.1016/j.vaccine.2021.05.069.
50. Nehal, K.R.; Steendam, L.M.; Campos Ponce, M.; van der Hoeven, M.; Smit, G.S.A. Worldwide Vaccination Willingness for COVID-19: A Systematic Review and Meta-Analysis. Vaccines 2021, 9, 1071, doi:10.3390/vaccines9101071.
51. Rodríguez-Blanco, N.; Montero-Navarro, S.; Botella-Rico, J.M.; Felipe-Gómez, A.J.; Sánchez-Más, J.; Tuells, J. Willingness to Be Vaccinated against COVID-19 in Spain before the Start of Vaccination: A Cross-Sectional Study. International Journal of Environmental Research and Public Health 2021, 18, 5272, doi:10.3390/ijerph18105272.
52. Syan, S.K.; Gohari, M.R.; Levitt, E.E.; Belisario, K.; Gillard, J.; DeJesus, J.; MacKillop, J. COVID-19 Vaccine Perceptions and Differences by Sex, Age, and Education in 1,367 Community Adults in Ontario. Frontiers in Public Health 2021, 9.
53. Wang, B.; Nolan, R.; Marshall, H. COVID-19 Immunisation, Willingness to Be Vaccinated and Vaccination Strategies to Improve Vaccine Uptake in Australia. Vaccines 2021, 9, 1467, doi:10.3390/vaccines9121467.
54. Szaszi, B.; Hajdu, N.; Szecsi, P.; Tipton, E.; Aczel, B. A Machine Learning Analysis of the Relationship of Demographics and Social Gathering Attendance from 41 Countries during Pandemic. Sci Rep 2022, 12, 724, doi:10.1038/s41598-021-04305-5.
“In the introduction, the authors describe their research and subsequent steps taken. They should, however describe in more detail what exactly was compared in the subsequent phases of the pandemic (lines 69-70). Failure to list individual elements raises the question of why the comparison was made at the very beginning and not all possible data presented?”
We have revised the text to address the Reviewer’s concerns and hope that the modified section in the Introduction is now more informative and convincing for the readers:
The following sentences have been modified and added to the Introduction section:
“Our study has several linked components. First, we describe the morbidity and mortality of the third pandemic wave, and we assess the spatial association between socio-economic characteristics of the population and morbidity, mortality due to COVID-19. Then we compare the second and third pandemic waves concerning the spatial distribution of the rates of confirmed COVID-19 cases and deaths. We, then, analyze the spatial distribution of vaccination coverage and its association with deprivation. Finally, we explore the interaction between vaccination coverage, mortality, and deprivation at the end of the third wave. In this way, we can look at the health consequences of relying primarily on mass vaccination, which had the benefit of speed but did not take account of the challenge of addressing inequalities.” (Lines 75-84 in the revised document).
“Line 118 - 2011 data are out of date, so why did the authors relied on them?”
We agree with the Reviewer that data used for deprivation index calculation were collected long ago, but most of the data for the calculation are only available from censuses, the last of which was in 2011 in Hungary. The next census is scheduled for the period between October 1 and November 20, 2022, and data processing will be completed by November 28, 2023. Anyhow, this is indeed a limitation, thus in the Discussion section, the following is mentioned as a limitation of the study:
“Another limitation of the study is the limited timeliness of the deprivation index. Most of the indicators used for the index calculation are only available from the census 2011 database and therefore, as for all such area-based socio-economic indicators, the accuracy of the index diminishes over time. It should be noted, however, that the spatial distribution of the indices based on the 2001 and 2011 census data, compiled with the same methodology (although not directly comparable in terms of values), did not show any significant, marked change and the deprived areas showed a nearly identical spatial distribution within Hungary, which suggests that the magnitude of differences in socioeconomic characteristics at territorial level is not changing, or if so, only very slowly and moderately. [20]” (Lines 427-435 in the revised document.)
Added literatures:
- Juhász, A.; Nagy, C.; Páldy, A.; Beale, L. Development of a Deprivation Index and Its Relation to Premature Mortality Due to Diseases of the Circulatory System in Hungary, 1998-2004. Soc Sci Med 2010, 70, 1342–1349, doi:10.1016/j.socscimed.2010.01.024.
“Line 196-197 – are the 5 cases described below 1 year and between 14-16? If so then this detail should be in parentheses. Otherwise the information is incomprehensible. Currently, it shows that, in addition to 5 people under the age of 18, also 2 died under 1 and 3 between 14-16.”
Thank you for pointing to this incomprehensive information. The sentence in lines 196-197 of the Results has been modified as follows:
“A total of five deaths were reported among those under 18 years (2 under 1 year and 3 aged between 14-16 years).” (Lines 205-207 in the revised document.)
“Information about a higher incidence rate among the Roma community is predictable. And it certainly results, among other things, from the way of life of this community. It is interesting, however, that the mortality rate among women in this social group is higher. However, there is no information to what extent these differences are statistically significant.”
We thank the Reviewer for pointing this out. Our results suggest that the point estimate of the relative mortality ratio is indeed higher for females than males, but it should be noted that the standard age-specific rates applied are different for males and females. Because of the different reference rates, the relative mortality ratio for females is not directly comparable for males. However, it is reasonable to suggest, that, considering 95% confidence intervals around the point estimates, the magnitude of the deviation from the sex-specific references, which indicates the degree of inequality, does not differ significantly between men and women.
We also note that when standardized death rates (European Standard Population [ESP] 2013.) were calculated, males had statistically significantly higher mortality rates than females (Males: 267.58 per 100000 persons; Females: 189.01 per 100 000 persons).
The following section has been added to the Results section to reflect on this finding:
“Concerning the districts with the highest representation of Roma population, a higher relative mortality ratio was identified for Roma females than for Roma males, but because of the different sex-specific reference rates, the relative mortality ratio for females is not directly comparable for males. However, considering 95% confidence intervals around the point estimates, the magnitude of the deviation from the sex-specific references, that indicates the degree of inequality, does not differ significantly between Roma men and women.” (Lines 400-406 in the revised document.)